# Dexmedetomidine use for patients in palliative care with intractable pain and delirium: A retrospective study

**Shi-Yuan Yu** [1]*, **Jacqueline Schellenberg**[2] , **Alison Alleyne**[3]

1 Department of Pharmacy, Burnaby Hospital, Burnaby, British Columbia, Canada, 2 Department of Pharmacy, Abbotsford Regional Hospital and Cancer Centre, Abbotsford, British Columbia, Canada, 3 Lower Mainland Pharmacy Services, Fraser Health, Langley, British Columbia, Canada

ʘ These authors contributed equally to this work.
* shiyuan.yu2@fraserhealth.ca

**Data Availability Statement:** All relevant data are within the manuscript and its Supporting Information files.

**Funding:** The author(s) received no specific funding for this work.

## Abstract

Patients seen by the palliative care team often have difficult and intractable symptoms. The current standard of practice to manage these symptoms is the deeply sedating midazolam continuous subcutaneous infusion for patients who are expected to expire within hours to days. Dexmedetomidine provides sedation but lacks evidence in palliative care use. This study describes continuous subcutaneous infusion of dexmedetomidine's effect on refractory pain and delirium. Retrospective, observational chart review and conducted in accordance with SQUIRE (quality improvement study). Twenty adult patients (18 years of age or older) with metastatic cancer disease admitted to three palliative complex care units of Fraser Health who received continuous subcutaneous infusion of dexmedetomidine between January 2017 to August 31, 2019. Average length of dexmedetomidine use was 9 days (1/3 length of stay). Eight of the 13 patients with pain symptoms exhibited an overall decline in pain. Four of the 6 patients with delirium had an initial decrease in delirium, but it did not last beyond the first day. Despite progressive clinical deterioration, adjunctive medications decreased or remained the same for 53% of as needed medications and 65% for regularly scheduled medications. Forty-five percent of patients had ≥50% days of rousable sedation. Hypotension occurred in 85% of patients. Dexmedetomidine provided benefit in managing intractable pain while allowing patients to remain rousable, but only had a short effect on delirium symptoms.

## Introduction

Patients nearing the end of life and suffering from a terminal illness are often transferred to an adult tertiary palliative care complex unit (PCCU) in Fraser Health (FH) for management of intractable symptoms such as pain and delirium. Palliative sedation becomes indicated when symptoms are refractory to usual management, and a patient is expected to expire within hours to days [1]. It is intended to lower the patient's level of consciousness during their last days of life [1]. Continuous subcutaneous infusion (CSCI) of midazolam is deeply sedating,

**Competing interests:** The authors have declared that no competing interests exist.

resulting in a patient's Richmond Agitation Sedation Scale (RASS), a validated assessment tool used to assess patient alertness and agitation, to be at least -4 [2]. CSCI midazolam is the recommended medication for palliative sedation, but often results in patients unable to have meaningful social interactions [1,3]. Other palliative sedation agents are less preferred due to cost, adverse effects, and administration logistics.

Dexmedetomidine is a selective agonist that binds to presynaptic $\alpha_2$-adrenoreceptors in the brainstem, creating a negative feedback loop that inhibits the release of norepinephrine and results in rousable anesthesia and sedation [4]. Dexmedetomidine has been shown to reduce pain, anxiety, and delirium without producing respiratory depression and can be given for hours to several days before a patient expires [4–10]. Dexmedetomidine use has been well established in critical care; however, published evidence for its use in palliative care is limited. Findings from case reports and cohort studies saw varying doses and administration methods used along with both objective and subjective pain and delirium assessments. A pilot study on its use in hyperactive delirium showed potential in managing refractory delirium, along with a recent review highlighting the prospective benefits of dexmedetomidine in palliative care [11,12]. In the FH PCCU a Palliative Performance Scale (PPS) is the validated tool used to assess physical status [13,14]. The Richmond Agitation-Sedation Scale (RASS) is another validated assessment tool used to assess patient alertness and agitation [2].

Limited published evidence and anecdotal experience in palliative care has led to interest in its use. This has resulted in a pre-printed order developed within the Fraser Health Authority to implement dexmedetomidine's use for these patients.

## Materials and methods

Retrospective, observational chart review of adult patients (18 years or older) admitted to a FH PCCU (Abbotsford Regional Hospital, Burnaby Hospital, and Surrey Memorial Hospital) who received CSCI dexmedetomidine between January 2017 to August 31, 2019. Fraser Health Research Ethics Board approved the study (FHREB 2019–096) with study conducted from October 8, 2019 to March 31, 2020.

This project was categorized as a "quality improvement" study by the FH Research Ethics board, with the goal of increasing awareness and knowledge of the use of dexmedetomidine in palliative pain management; as such, consent from patient or family was not required.

Data collected included the following:

- Patient demographics (sites, age, sex, weight, medical condition(s)/reasons requiring palliative care, reason for admission to PCCU, length of palliative stay in days, reason for discontinuation of palliative stay, heart rate, blood pressure, respiratory rate, intractable symptoms of either pain or delirium present)

- Dexmedetomidine use (dosing, duration of use in days, reason for discontinuation);

- Adjunctive medication use (duration of its use prior and during CSCI dexmedetomidine, and its maximum dose used up to 24 hours after cessation of CSCI dexmedetomidine);

- Clinical outcomes:

   ○ Side effects of dexmedetomidine: hypotension(systolic blood pressure <90 mmHg), hypertension (systolic blood pressure >150 mmHg), bradycardia (heart rate < 50 beats per minutes), tachycardia (heart rate >110 beats per minute), bradypnea (respiratory rate <10 breaths per minute). These are collected as per nursing documentation while on dexmedetomidine.

○ RASS scores 24 hours before, daily during, and 24 hours after dexmedetomidine therapy

○ Symptoms of pain prior, during, and after discontinuation of dexmedetomidine therapy as noted by a numbered scale completed daily.

○ Symptoms of delirium prior, during, and after discontinuation of dexmedetomidine therapy as noted by a numbered scale completed daily.

Descriptive statistics were used to analyze patient demographics and the objectives of our study. For baseline patient demographics, means with standard deviations and medians with interquartile ranges were provided for normally and non-normally distributed values, respectively.

For clinical outcomes, a reduction in pain or delirium was characterized as any reduction from their initial pain or delirium score prior to the initiation of dexmedetomidine. As a multidisciplinary team within the palliative care unit, these scores were collected from nursing charts or through physician's progress notes.

## Results and discussion

Twenty patients fulfilled the inclusion criteria. The average age of the patients was 49.5 (SD ± 13.3) years old, and all had metastatic cancer. Detailed baseline characteristics and reasons for initiation of dexmedetomidine are described in **Table 1**. Average length of stay was 28 days (SD ± 12.2) and 95% of patients expired during their admission.

Intractable pain was the reason for starting dexmedetomidine in 90% of patients. The use of dexmedetomidine is detailed in **Table 2**. It started 15.9 days (± 12.5) after admission to the PCCU, with no initial bolus dose. The infusion was titrated from 0.19 (± 0.05) to 1.07 (± 0.29) mcg/kg/h, for a mean duration of 8.7 (± 4.8) days, with 80% of patients continuing until death and 15% discontinuing due to side effects.

The effects of dexmedetomidine on pain are illustrated in **Fig 1**. **Fig 1** shows pain scores for patients treated up to 7 days (n = 6), and for those treated up to 14 days (n = 7). Eight patients showed an overall decline in pain scores while on dexmedetomidine, with benefit seen in both short- and long-term groups. For delirium, only the patients who had this intractable symptom as one of the reasons for starting dexmedetomidine were analyzed. No validated delirium scoring system was documented and therefore each delirious symptom noted in the nursing chart were coded as one point, with a maximum score of 5 points total. Four of 6 patients had a decrease in delirium scores on the day dexmedetomidine was started, but the benefit did not last beyond day 1 for 3 of those 4 patients. Midazolam CSCI was added to 4 patients to reduce their delirium symptoms.

A gradual decline in PPS scores was seen in all patients. The two most common side effects were hypotension (17 of 20 patients; 2.1 ± 1.3 episodes per patient; beginning 40 ± 59.1 h after initiation) and tachycardia (7 patients; 1.5 ± 1.3 episodes; beginning 46.3 ± 78.7 h after initiation). Nine (45%) patients experienced 50% or more days with a RASS score of 0 to -2 (alert to light sedation) while on dexmedetomidine [2].

Many other adjunctive agents were used prior to the initiation of dexmedetomidine. These included opioids, ketamine, lidocaine, phenobarbital, haloperidol, methotrimeprazine, midazolam, and benzodiazepines. After dexmedetomidine was initiated, 53% and 65% of "as needed" and "regularly scheduled" adjunctive medication use was decreased or unchanged, respectively, despite declining patient clinical status. This showed that dexmedetomidine use could spare an increase in the number of adjunctive medications used and was in line with the patients' wishes to remain more awake at the end of life. Midazolam CSCI was initiated in 55% of the patients while on dexmedetomidine. The overall usage of various regularly scheduled

**Table 1. Baseline characteristics.**

|  | N = 20 |
|---|---|
| Age, y (mean ± SD) | 49.5 (± 13.3) |
| Sex, female (%) | 60 |
| Site (%) |  |
| ARH | 65 |
| BH | 30 |
| SMH | 5 |
| Weight, kg (mean ± SD) | 59 (± 15.9) |
| LOS, days (mean ± SD) | 28 (± 12.2) |
| Vitals on admission |  |
| Heart rate (mean ± SD) | 100 (± 17.7) |
| SBP (mean ± SD) | 114 (± 21.1) |
| DBP (mean ± SD) | 69 (± 16.9) |
| PPS on admission (mode) | 40 |
| Vitals prior to dexmedetomidine |  |
| Heart rate (mean ± SD) | 106 (±13.3) |
| SBP (mean ± SD) | 119 (±25.8) |
| DBP (mean ± SD) | 74 (±12.1) |
| PPS prior to dexmedetomidine (mode) | 30 |
| Medical Condition (%) |  |
| Metastatic gastrointestinal cancer | 30 |
| Metastatic lung cancer | 25 |
| Metastatic gynecological cancer | 25 |
| Metastatic sarcoma | 15 |
| Metastatic genitourinary cancer | 5 |
| Reason for admission to TPCU (%) |  |
| Pain | 85 |
| Delirium | 15 |
| Dyspnea | 5 |
| Poor oral intake | 5 |
| Intractable symptom for initiating dexmedetomidine (%) |  |
| Pain | 90 |
| Delirium | 30 |
| Dyspnea | 25 |

**Table 2. Characteristics of dexmedetomidine use.**

|  | N = 20 |
|---|---|
| Duration in PCCU before dexmedetomidine started, days (mean ± SD) | 15.9 (± 12.5) |
| Length of use, days (mean ± SD) | 8.7 (± 4.8) |
| Dose, mcg/kg/hr (mean ± SD) |  |
| Initial | 0.19 (± 0.05) |
| Maximum | 1.07 (± 0.29) |
| End | 0.86 (± 0.44) |
| Cause of discontinuation (%) |  |
| Death | 80 |
| Failure to manage intractable symptoms | 5 |
| Other (side effects) | 15 |

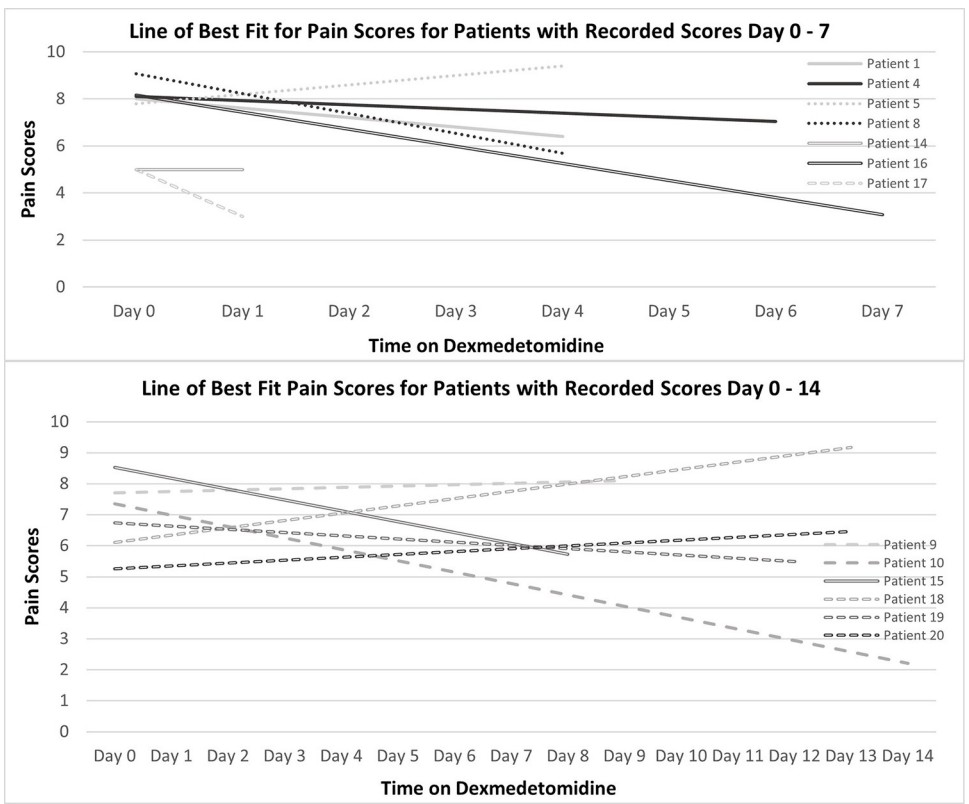

**Fig 1. Line of best fit for pain scores of patients with recorded scores day 0–7 and day 0–14.**

and as-needed (PRN) adjunctive medications during dexmedetomidine use when compared to 24h after dexmedetomidine discontinuation are described in **Table 3**.

The mean duration of use of dexmedetomidine was 8.7 days (± 4.8), about 1/3 of their time on the unit. This may be due to its clinical benefits, while minimizing the need for increased adjunctive medications. Dexmedetomidine was discontinued for one patient, due to no perceived benefit; and for 3 patients with intolerable side effects (2 hypotension, 1 dyspnea) as documented in the patient's charts. No loading doses were used; which differs from some case reports [15,16]. No patient exceeded the maximum ICU dexmedetomidine infusion rate of 1.4 mcg/kg/h, which was consistent with published case reports in adults. Recommendations from a recent review article suggest a loading dose may be possible but many clinicians refrain from ordering it [12]. Further exploration on dose optimization and use of a loading dose would be beneficial.

Most patients experienced delirium relief that only lasted the first day dexmedetomidine was started. This may be due to the low starting dose, but effectiveness of dexmedetomidine for delirium management is unclear. Two patients had midazolam CSCI added on day 2 and 3, which lowered their delirium symptom score; and it was later added to another 2 patients on days 9 and 11, respectively. It is unknown how much delirium management was due to midazolam versus dexmedetomidine CSCI.

Even though 18 patients had intractable pain, only 13 patients had documented pain scores to be analyzed. Eight of 13 patients showed an overall decline in pain scores. Midazolam CSCI was only added on the last day of recorded pain scores for patients 9 and 18 alongside increasing regularly scheduled methadone dose, as they had increasingly worsening pain symptoms

**Table 3.** Individual patient breakdown of overall adjunctive medication usage during dexmedetomidine use compared with 24h after dexmedetomidine discontinuation.

| | MEDICATIONS | OVERALL REGULAR MEDICATION USE* | OVERALL PRN MEDICATION USE* | OVERALL EFFECT4 | OVERALL FOR PATIENT5 |
|---|---|---|---|---|---|
| **Patient 1** | Methadone | S | N/A | S | **67% R,** 33% S |
| | Hydromorphone | S | R | R | |
| | Ketamine | S | R | R | |
| | Methotrimeprazine | N/A | R | R | |
| | Clonazepam | S | N/A | S | |
| | Lorazepam | N/A | R | R | |
| **Patient 2** | Methadone | I | N/A | I | **57% I,** 29% R, 14% S |
| | Hydromorphone | S | R | R | |
| | Dexamethasone | S | N/A | S | |
| | Ketamine | I | I | I | |
| | Methotrimeprazine | N/A | I | I | |
| | Midazolam | I | I | I | |
| | Lorazepam | N/A | R | R | |
| **Patient 3** | Methadone | S | N/A | S | 37.5 S, 37.5% I, 25% R |
| | Hydromorphone | I | R | R | |
| | Dexamethasone | S | N/A | S | |
| | Ketamine | I | R | I | |
| | Haloperidol | N/A | I | I | |
| | Methotrimeprazine | N/A | S | S | |
| | Midazolam | I | I | I | |
| | Lorazepam | N/A | R | R | |
| **Patient 4** | Methadone | I | N/A | I | **57% I,** 29% R, 14% S |
| | Hydromorphone | N/A | R | R | |
| | Dexamethasone | S | N/A | S | |
| | Ketamine | R | I | R | |
| | Methotrimeprazine | I | I | I | |
| | Midazolam | I | I | I | |
| | Lorazepam | N/A | I | I | |
| **Patient 5** | Methadone | S | N/A | S | 43% R, 43% S, 14% I |
| | Hydromorphone | S | R | R | |
| | Dexamethasone | S | N/A | S | |
| | Ketamine | S | R | R | |
| | Methotrimeprazine | S | S | S | |
| | Midazolam | I | I | I | |
| | Lorazepam | N/A | R | R | |
| **Patient 6** | Hydromorphone | I | S | I | **100% I** |
| | Dexamethasone | I | N/A | I | |
| | Methotrimeprazine | I | I | I | |
| | Midazolam | I | I | I | |
| **Patient 7** | Methadone | I | N/A | I | **75% I,** 25% S |
| | Hydromorphone | I | R | I | |
| | Dexamethasone | S | N/A | S | |
| | Phenobarbital | I | I | I | |
| | Haloperidol | S | I | I | |
| | Methotrimeprazine | I | I | I | |
| | Midazolam | I | I | I | |

*(Continued)*

**Table 3.** (*Continued*)

| | MEDICATIONS | OVERALL REGULAR MEDICATION USE* | OVERALL PRN MEDICATION USE* | OVERALL EFFECT4 | OVERALL FOR PATIENT5 |
|---|---|---|---|---|---|
| | Clonazepam | S | N/A | S | |
| Patient 8 | Methadone | S | N/A | S | **50% S**, 25% R, 25% I |
| | Sufentanil | N/A | I | I | |
| | Hydromorphone | N/A | R | R | |
| | Dexamethasone | S | N/A | S | |
| Patient 9 | Methadone | I | N/A | I | **83% I**, 17% S |
| | Hydromorphone | N/A | I | I | |
| | Fentanyl Patch | S | N/A | S | |
| | Ketamine | I | R | I | |
| | Midazolam | I | I | I | |
| | Lorazepam | N/A | I | I | |
| Patient 10 | Methadone | I | N/A | I | **57% I**, 29% R, 14% S |
| | Fentanyl | I | R | R | |
| | Dexamethasone | I | N/A | I | |
| | Ketamine | I | N/A | I | |
| | Haloperidol | I | N/A | I | |
| | Midazolam | N/A | R | R | |
| | Lorazepam | N/A | S | S | |
| Patient 11 | Methadone | S | N/A | S | 43% S, 43% I, 14% R |
| | Fentanyl | R | I | R | |
| | Hydromorphone | N/A | I | I | |
| | Dexamethasone | S | N/A | S | |
| | Phenobarbital | I | N/A | I | |
| | Haloperidol | S | N/A | S | |
| | Midazolam | I | I | I | |
| Patient 12 | Hydromorphone | S | S | S | **43% I**, 28.5% R, 28.5% S |
| | Dexamethasone | S | N/A | S | |
| | Phenobarbital | N/A | I | I | |
| | Methotrimeprazine | R | R | R | |
| | Midazolam | R | S | R | |
| | Lorazepam | N/A | I | I | |
| | Loxapine | I | I | I | |
| Patient 13 | Methadone | S | N/A | S | **50% S**, 33% R, 17% I |
| | Hydromorphone | N/A | R | R | |
| | Dexamethasone | S | N/A | S | |
| | Haloperidol | S | N/A | S | |
| | Midazolam | N/A | I | I | |
| | Lorazepam | N/A | R | R | |
| Patient 14 | Methadone | I | I | I | 43% S, 43% I, 14% R |
| | Hydromorphone | S | N/A | S | |
| | Fentanyl | I | I | I | |
| | Dexamethasone | S | N/A | S | |

(*Continued*)

**Table 3.** (Continued)

| | MEDICATIONS | OVERALL REGULAR MEDICATION USE* | OVERALL PRN MEDICATION USE* | OVERALL EFFECT4 | OVERALL FOR PATIENT5 |
|---|---|---|---|---|---|
| | Haloperidol | N/A | S | S | |
| | Methotrimeprazine | I | I | I | |
| | Midazolam | R | R | R | |
| Patient 15 | Methadone | S | R | R | **50% I,** 25% R, 25% S |
| | Hydromorphone | I | S | I | |
| | Dexamethasone | S | N/A | S | |
| | Lorazepam | N/A | I | I | |
| Patient 16 | Methadone | S | N/A | S | **43% R,** 28.5% S, 28.5% I |
| | Sufentanil | N/A | R | R | |
| | Dexamethasone | R | N/A | R | |
| | Haloperidol | S | N/A | S | |
| | Methotrimeprazine | I | I | I | |
| | Midazolam | I | I | I | |
| | Lorazepam | N/A | R | R | |
| Patient 17 | Methadone | S | N/A | S | **40% R,** 40% S, 20% I |
| | Hydromorphone | N/A | R | R | |
| | Fentanyl | S | N/A | S | |
| | Methotrimeprazine | S | R | R | |
| | Midazolam | I | I | I | |
| Patient 18 | Methadone | I | N/A | I | **57% I,** 43% R |
| | Hydromorphone | N/A | R | R | |
| | Dexamethasone | I | N/A | I | |
| | Ketamine | I | I | I | |
| | Methotrimeprazine | N/A | R | R | |
| | Midazolam | I | I | I | |
| | Lorazepam | N/A | R | R | |
| Patient 19 | Methadone | S | R | R | **56% I,** 22% R, 22% S |
| | Hydromorphone | N/A | R | R | |
| | Dexamethasone | S | N/A | S | |
| | Ketamine | I | R | I | |
| | Haloperidol | N/A | I | I | |
| | Methotrimeprazine | N/A | I | I | |
| | Midazolam | I | I | I | |
| | Clonazapam | S | N/A | S | |
| | Lorazepam | N/A | I | I | |
| Patient 20 | Methadone | S | N/A | S | **67% I,** 16.5% R, 16.5% S |
| | Fentanyl | I | R | I | |
| | Ketamine | I | I | I | |
| | Methotrimeprazine | N/A | I | I | |
| | Midazolam | N/A | R | R | |
| | Clonazapam | I | N/A | I | |

*S = same, R = reduced, I = increased.

and died 3 and 2 days later, respectively. The ability of these patients to provide pain scores demonstrated that dexmedetomidine managed intractable pain for quite a few days and they were rousable enough to communicate with the nursing staff. This also suggests that dexmedetomidine may be "midazolam-sparing" for some patients.

The decline in PPS scores reflects symptom and disease progression but it is unknown if dexmedetomidine slowed the rate of decline. A comparison against a group of patients who did not receive dexmedetomidine could be further explored. Although the RASS-PAL [17] would be the preferred sedation assessment tool for palliative patients, the RASS was used instead due to its familiarity locally. Having 45% of patients experiencing 50% or more days with a rousable RASS score of 0 to -2 while on dexmedetomidine showed it provided rousable sedation and improved intractable symptoms. It was noted that these patients had periods of alert and calm to easily roused when drowsy, facilitating their ability to still have contact with those close to them.

Cardiovascular and respiratory side effects are clearly associated with dexmedetomidine. However, only 15% of patients discontinued dexmedetomidine due to side effects as per the patients' requests. Often, most instances of side effects were resolved with slight dose tapering as initiated by the nurse. As there was no clinical protocol at the time of this study, if side effects were observed then a physician's subjective assessment and through discussion with the rest of the interprofessional team would the infusion then be weaned and discontinued.

Prior to the initiation of dexmedetomidine, other adjunctive medications were trialed to manage intractable symptoms. Despite their progressive clinical deterioration, the overall adjunctive medication use decreased or remained unchanged in 53% of as needed and 65% of regularly prescribed medications. Midazolam CSCI was initiated most frequently while on dexmedetomidine, in 55% of patients. The absence of initiating midazolam CSCI use in 9 patients suggests that dexmedetomidine can help provide the rousable sedation and comfort towards end of life.

Dexmedetomidine use may have financial implications as it is approximately twenty times more expensive than midazolam during a 24-hour infusion period at our sites. Recent patent expiration may lead to more competitive pricing.

A major limitation of this study was its retrospective observational cohort study design; data extraction was limited to what was available within the charts. Many symptom data points were not documented or were noted subjectively by healthcare professionals, requiring investigator interpretation of data points. Clinical decisions regarding the initiation of dexmedetomidine may not have been completely documented and/or may have been misrepresented by the investigator. The number of patients is considered large for a palliative study but with only 20 patients it is still subject to significant individual variabilities.

Further directions for research include prospectively comparing dexmedetomidine CSCI and midazolam CSCI to evaluate symptom management, adjunctive medication usage, PPS scores, and RASS-PAL. Having objective data consistently documented ensures that outcomes will be accurately assessed. As well, the RASS-PAL should be used in the scoring of rousable sedation. Since the completion of this study, a Fraser Health pre-printed order has been created guiding the initiation and titration of CSCI dexmedetomidine along with requirements of objective data to be documented which can help future studies.

## Conclusion

Dexmedetomidine can provide benefits in managing intractable pain symptoms while allowing palliative patients to maintain rousable sedation. It may also provide some initial delirium relief. There was some stable or decreased use in adjunctive medication usage. Further

research is warranted to determine the extent of benefit from dexmedetomidine CSCI, and its place in palliative care treatment guidelines.

## Author Contributions

**Conceptualization:** Shi-Yuan Yu, Jacqueline Schellenberg, Alison Alleyne.

**Data curation:** Shi-Yuan Yu.

**Methodology:** Shi-Yuan Yu, Jacqueline Schellenberg, Alison Alleyne.

**Supervision:** Jacqueline Schellenberg, Alison Alleyne.

**Writing – original draft:** Shi-Yuan Yu.

**Writing – review & editing:** Shi-Yuan Yu, Jacqueline Schellenberg, Alison Alleyne.

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
