## [Decision Letter · Decision Letter 0]

16 Apr 2023

PONE-D-23-07176Dexmedetomidine use for palliative care patients with intractable pain and delirium: A retrospective reviewPLOS ONE

Dear Dr. Yu,

Thank you for submitting your manuscript to PLOS ONE. After careful consideration, we feel that it has merit but does not fully meet PLOS ONE’s publication criteria as it currently stands. Therefore, we invite you to submit a revised version of the manuscript that addresses the points raised during the review process.

We look forward to receiving your revised manuscript.

Kind regards,

Tommaso Martino, M.D.

Academic Editor

PLOS ONE

Journal Requirements:

Additional Editor Comments: 

Please find the comments of our three referees.

Reviewers' comments:

Reviewer's Responses to Questions

**Comments to the Author**

1. Is the manuscript technically sound, and do the data support the conclusions?

Reviewer #1: Yes

Reviewer #2: Yes

Reviewer #3: Yes

2. Has the statistical analysis been performed appropriately and rigorously? 

Reviewer #1: N/A

Reviewer #2: N/A

Reviewer #3: Yes

3. Have the authors made all data underlying the findings in their manuscript fully available?

Reviewer #1: Yes

Reviewer #2: Yes

Reviewer #3: Yes

4. Is the manuscript presented in an intelligible fashion and written in standard English?

Reviewer #1: Yes

Reviewer #2: Yes

Reviewer #3: Yes

5. Review Comments to the Author

Reviewer #1: This is a retrospective review of the use of dexmedetomidine in the treatment of pain and delirium for patients in an inpatient palliative care unit setting. This authors find mixed results in the palliation of those symptoms with some diminution in effect over the course of treatment though accompanied by less utilization of as-needed medication dosing. My overall concerns are as follows:

(1) Terms throughout need operative definitions: "deeply sedating" "proportional rousable sedation" "palliative sedation" need to be defined as objectively as possible to ease comprehensiveness

(2)Avoid euphemisms such as "pass away," which also comes from a Christian context and is therefore exclusive of other faith traditions

(3) Would use person-first language (https://www.cdc.gov/ncbddd/disabilityandhealth/materials/factsheets/fs-communicating-with-people.html) throughout and avoid "palliative care patients" and rather defining them as "patients/people being seen by the palliative care team"

(4) Within the methods section, I think much of what you have included in the supplementary materials needs to be included here including more detail about how many pain and delirium assessments were included (all of them or just certain ones during the day); it would also be useful to understand how BP was collected (were all measured BPs collected and averaged? What is the protocol in these units for vital sign collection on the dying? Were there patients who didn't have vital signs collected?), how was delirium assessed, this isn't clear in the manuscript or the supplement; how were pain scores collected for those who can self-report vs. those who could not? This section also needs operative definition for how the authors determined: hypotension (and why was SBP chosen rather than MAP; how was it determined that hypotension was "symptomatic" was this only in those who could verbally report?), reduction in pain (most literature defines a 30% reduction in NPS as clinically meaningful), and reduction in delirium. Were these pre-defined or developed ad hoc?

(5) Within the methods it would also be helpful to understand what the palliative care teams look like at these sites. Are they multidisciplinary/intraprofessional?

(6) Within the results section, it would be helpful to have an idea of what sorts of interventions had been tried and failed before dexamethasone initiation. This would help other teams understand when to consider using it.

(7) In the results section, I recommend that Tables S1 and S2, be moved into the body of the manuscript rather than the supplement.

(8) Do these sites have one or more protocols for use of dexamethasone that guide initiation and titration? If so, did they follow them? If so, can they be shared in the supplement for other teams to see and use?

(9) In the results, inclusion of in-line medians and StDev would be useful for things like pain scores, delirium scores, and RASS to help the reader follow along.

(10) Within results, it would be helpful to understand if those who were more lightly sedated also required more as needed medications? Was it actually a good thing that they were more awake?

(11) It would be useful to understand more objectively what was happening with the opioids during this time. Can MEDD/OME be included (and, if so, I will also ask that the conversion factors that are used to calculate MEDD be included in the supplement as well to help with reproducibility)? This is particularly of interest to me for patients 9 & 18 where the discussion section mentions that midazolam was started for pain, which seems concerning if the opioids weren't also being adjusted.

(12) Line 143-144 of the discussion introduces new information that the dexmedetomidine was weaned for side effects. Would like to see more about this in the results section. When was that considered? How was it weaned? Is this part of a clinical protocol?

(13) I'm not sure that the separate line of best fit figures are necessary. I think the 0-14 day figure would be sufficient if there is more context in the body of the manuscript re: median pain scores.

(14) If the authors find it relevant, I would be honored to see our case report on dexmedetomidine at EOL also be included as a reference (https://doi.org/10.1080/15360288.2022.2102705).

This is a very important, leading-edge option for palliation and I am excited to see this manuscript. I hope the above recommendations are taken in that spirit of excited anticipation.

Reviewer #2: This paper presents a case serie for palliative care patients treated with dexmedetomidine (DEX). This use of DEX is promising and at his stage it is good to learn from initial clinical experience. The paper is well written and the discussion reflect the inherent limitations in a retrospective analysis.

Some comments:

1. The authors state that the use of DEX for 8.7 days is longer that in other reports. About at the same time this manuscript was submitted we published a case serie with treatment intervals of median 40 days. Obviously this paper was not available to the authors at the time they prepared their manuscript, but perhaps this information could be added in a revised version. Our paper is available for free at

https://www.ncbi.nlm.nih.gov/pmc/articles/PMC10036681/pdf/40122_2023_Article_485.pdf

2. I miss the development of opioid dose after start DEX; decline, stable or increase? This is needed in order to interpret the potential analgesic benefit from DEX where a part of the success would be to decrease opioids.

3. In the introduction is is correctly argued that DEX may have the patients more arousable than similar sedation with midazolam. However, I miss information about to what extent the patients in this case serie after start of DEX was able to have some contact with relatives etc.

4.Hypotension is not defined in the paper. It may be in the supplementary, but let's face it nobody actually read supplementary. The important question is if the hypotension was clinical significant. In our experience it will not be a clinical issue and in this patents setting why really focus on such effects.

6. page 5 line 95. Small but important typo, not mg but mikrogram

Reviewer #3: Thank you for letting me review this paper on the use of dexmedetomidine in palliative care. The paper is well written and reports on a retrospective chart review of 20 relatively young patients with metastatic cancer disease receiving sc dexmedetomidine due to intractable symptoms in end of life.

The format is more of a short report than an original article. The paper adds to the current scanty literature on the use of dexmedetomidine in palliative care.

My specific comments:

- Abstract, background: Please stress that deep sedation with midazolam in palliative care is limited to end-of-life care.

- Abstract, setting: According to Table S1 all patients had metastatic cancer disease. Adding this information in the text helps to clarify the patient group studied.

- Introduction, third paragraph: I lack some newer references, e.g. the article by Lohre et al in Pain Ther 2023 and the review by Gaertner in Ann Palliat Med 2022 which provides practice recommendations.

- Methods, third paragraph: I question the detailed description of the data collection instrument and subsequent security routines. Better to replace this with a comprehensive description of the symptom assessment instruments used in the study (mentioned in Introduction and described in supplemental data) together with a short description of statistical methods used (now only described in supplemental data).

- Results, overall: Both Table S1 and S2 provides essential data that is important for the paper. I suggest having them in the paper instead of in supplemental data.

- Results, first paragraph: Adding information that all patients had metastatic cancer and the mean age helps the reader understand the patient group studied.

- Results, second paragraph: The authors describe intractable pain as the main reason for starting dexmedetomidine. The paper would benefit from, if possible, having the pain mechanism(s) described for the patients to better understand the different treatment responses observed.

- Results, third paragraph: It is unclear to me how delirium was assessed and scored, please add this information to Methods. Also, figure 2 is “blurry” and difficult to read. I suggest excluding this figure but keeping the described results in the text.

- Results, fourth paragraph: Hypotension and tachycardia were described as the two most common side effects of the treatment. Parallel to this, a gradual decline in PPS scores were seen. Hypotension per se is common in end of life. I lack a discussion around this in Discussion – how could you distinguish treatment related hypotension from the expected hypotension in these dying patients?

- Discussion, second paragraph, third line: Please correct the sentence.

- Discussion, sixth paragraph, fourth line: The authors present data about midazolam CSCI that is not presented in Results. Please add this information to Results, enabling the following discussion.

6. PLOS authors have the option to publish the peer review history of their article (what does this mean?). If published, this will include your full peer review and any attached files.

Reviewer #1: **Yes: **Kyle P. Edmonds, MD FAAHPM

Reviewer #2: **Yes: **Pål Klepstad

Reviewer #3: No

---

## [Author Response · Author response to Decision Letter 0]

10 Jul 2023

See attached "response to reviewers"

---

## [Decision Letter · Decision Letter 1]

11 Sep 2023

Dexmedetomidine use for patients in palliative care with intractable pain and delirium: A retrospective study

PONE-D-23-07176R1

Dear Dr. Yu,

We’re pleased to inform you that your manuscript has been judged scientifically suitable for publication and will be formally accepted for publication once it meets all outstanding technical requirements.

Kind regards,

Steve Zimmerman, PhD

Associate Editor, PLOS ONE

Additional Editor Comments (optional):

Reviewers' comments:

Reviewer's Responses to Questions

**Comments to the Author**

1. If the authors have adequately addressed your comments raised in a previous round of review and you feel that this manuscript is now acceptable for publication, you may indicate that here to bypass the “Comments to the Author” section, enter your conflict of interest statement in the “Confidential to Editor” section, and submit your "Accept" recommendation.

Reviewer #1: All comments have been addressed

Reviewer #2: All comments have been addressed

Reviewer #3: All comments have been addressed

2. Is the manuscript technically sound, and do the data support the conclusions?

Reviewer #1: Yes

Reviewer #2: Yes

Reviewer #3: Yes

3. Has the statistical analysis been performed appropriately and rigorously? 

Reviewer #1: Yes

Reviewer #2: Yes

Reviewer #3: Yes

4. Have the authors made all data underlying the findings in their manuscript fully available?

Reviewer #1: Yes

Reviewer #2: Yes

Reviewer #3: Yes

5. Is the manuscript presented in an intelligible fashion and written in standard English?

Reviewer #1: Yes

Reviewer #2: Yes

Reviewer #3: Yes

6. Review Comments to the Author

Reviewer #1: My gratitude to the authors for addressing the reviewers' comments so comprehensively. The manuscript is very sound and will make an important contribution to the literature. I have no further requested revisions.

Reviewer #2: Always potential for improvement, and perhaps the journal will supply editorial changes, however I find the revised version generally satisfactory

Reviewer #3: The authors have satisfactory addressed the comments raised by the reviewers and I have no further comments.

7. PLOS authors have the option to publish the peer review history of their article (what does this mean?). If published, this will include your full peer review and any attached files.

Reviewer #1: **Yes: **Kyle P. Edmonds, MD FAAHPM

Reviewer #2: **Yes: **Pål Klepstad

Reviewer #3: **Yes: **Staffan Lundström MD, PhD

---

## [Editor Report · Acceptance letter]

18 Sep 2023

PONE-D-23-07176R1 

Dexmedetomidine use for patients in palliative care with intractable pain and delirium: A retrospective study 

Dear Dr. Yu:

I'm pleased to inform you that your manuscript has been deemed suitable for publication in PLOS ONE. Congratulations! Your manuscript is now with our production department. 

Kind regards, 

on behalf of

Dr. Amir Radfar 

Academic Editor

PLOS ONE